# Multi-Mask Aggregators for Graph Neural Networks

**Ahmet Sarıgün**
Middle East Technical University
`ahmet.sarigun@metu.edu.tr`

**Ahmet S. Rifaioğlu**
Institute of Computational Biomedicine
Heidelberg University
`ahmet.rifaioglu@uni-heidelberg.de`

## Abstract

One of the most critical operations in graph neural networks (GNNs) is the aggregation operation, which aims to extract information from neighbors of the target node. Several convolution methods have been proposed, such as standard graph convolution (GCN), graph attention (GAT), and message passing (MPNN). In this study, we propose an aggregation method called Multi-Mask Aggregators (MMA), where the model learns a weighted mask for each aggregator before collecting neighboring messages. MMA draws similarities with the GAT and MPNN but has some theoretical and practical advantages. Intuitively, our framework is not limited by the number of heads from GAT and has more discriminative than an MPNN. The performance of MMA was compared with the well-known baseline methods in both node classification and graph regression tasks on widely-used benchmarking datasets, and it has shown improved performance. Dataset and codes are available at https://github.com/asarigun/mma.

## 1 Introduction

Graph Neural Networks (GNNs) have attracted great interest in recent years due to their performance and the ability to extract complex information [1–4]. One of the most critical operations in graph neural networks is the aggregation operation, where the aim is iteratively exploiting information from the neighbors of a target node to update its latent representation. [2, 5]. Several different aggregators were used, such as mean, sum, max, min, and long short-term memory (LSTM), to extract more meaningful information from the neighbors of a particular node. [4], [5]. According to [6], an ideal learnable and flexible aggregation should have the following conditions: 1) permutation invariant [7]; 2) adaptive to deal with various neighborhood information [3] [8]; 3) explainable learned representations concerning the predictions and robustness to the noise [9] 4) discriminative to graph structures [5].

Several methods have been proposed in the graph neural network area in recent years that use different aggregators. For example, Graph Attention Network (GAT) borrows the idea of attention mechanisms that perform aggregations by assigning different weights to different neighbors [3]. However, it is not adaptive to deal with various neighborhood information at the feature level since all individual features are considered equally [3] [8]. Learnable graph convolutional layer (LGCL) method applies convolution operation in the aggregation process by assigning different weights to different features [8].LGCL can deal with different neighborhood information; however, there might be loss information on graphs during the selections since it breaks the original correspondence between node features by selecting the d-largest feature values from the neighboring nodes [3] [8]. Dehmamy et al. [10] empirically showed that using multiple aggregators (i.e., mean, max, and normalized mean) improves the performance of GNNs on the task of graph moments. Principal Neighbourhood Aggregation (PNA) method theoretically formalized this observation. The authors demonstrated that using a single type of aggregator is insufficient to extract enough information from neighboring nodes which causes limited learning abilities and expressive power [11].

A mask aggregator uses an auxiliary model such as multi-layer perceptrons (MLPs), which has no requirement for size or order of the input datasets [12] [13]. To satisfy the four conditions mentioned

A. Sarıgün et al., Multi-Mask Aggregators for Graph Neural Networks (Extended Abstract). Presented at the First Learning on Graphs Conference (LoG 2022), Virtual Event, December 9–12, 2022.

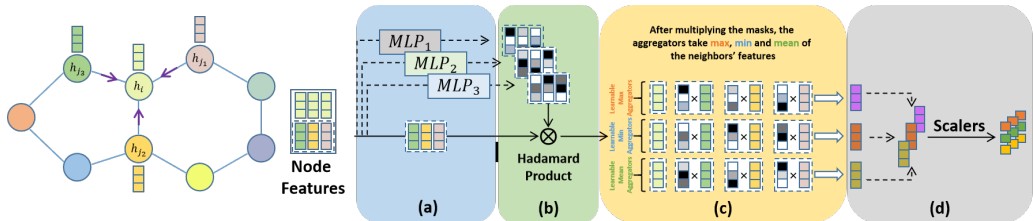

**Figure 1:** Architecture of MMA with the different aggregators: a) training auxiliary model with a given node and its neighbors' feature vectors; b) getting the masks for each neighbor from the auxiliary model and multiplying with Hadamard product with node feature and Learned Mask; c) aggregating the neighbors (after multiplying the corresponding mask) to get the central node's new representation. d) combining the final aggregators with scalers.

above, Learnable Aggregator for GCN (LA-GCN) was proposed, which filters the neighborhood information with a mask aggregator before the aggregation process [6]. LA-GCN learns a specific mask for each node's neighbor, allowing node-level and feature-level attention by the auxiliary model. This mechanism assigns different weights to nodes and features, providing interpretable results and increasing the model's robustness. However, LA-GCN is only based on a sum aggregator, which loses its stability with the increasing average degree of a graph [11], and the other types of aggregators are overlooked.

In this study, we propose Multi-Mask Aggregators (MMA), a novel graph neural network method that combines trainable auxiliary models with different or the same aggregators. MMA utilizes a given node and its neighbors to train auxiliary models to extract information in a graph where different neighborhood information is learned using different masks. We use multiple types of aggregators (i.e., mean, max, and min) and create a mask for each neighbor and aggregator. MMA can learn high-level rules (e.g., focusing on the important neighbors and features for node representation learning) to guide the aggregators for better utilization of the neighborhood information. It is a flexible method where different or the same kinds of multiple learnable aggregators can be used. We evaluated MMA on well-known benchmark datasets and compared its performance with the well-known baseline graph neural network methods. The datasets, source code, experimental settings, and user instructions are available publicly at `https://github.com/asarigun/mma`.

The main contributions can be summarized as the followings: **1)** It provides **flexible** multi-aggregators with the mask aggregation;**2)** It **unlocks** the limitation on the number of heads; **3)** It enables to extract **local information** by local parameters instead of using global parameters like in MPNNs/PNA; **4)** It behaves between in **GAT** and **MPNN/PNA**; **5)** It increases the performance in **node classification** and **graph regression** benchmarks.

## 2 Multi-Mask Aggregators

The proposed Multi-Mask Aggregators method leverages the increased expressivity from the multi-aggregators models such as PNA [11], and the learnable masks from LA-GCN [6]. The Hadamard product is performed to multiply the neighbor's feature vector with the corresponding learned masks in the aggregation process, allowing each heuristic aggregator (e.g., min, mean, etc.) to learn different features from the neighbors. Finally, the resulting aggregators are combined with scalers [11]. MMA architecture is given in Figure 1.

### 2.1 Motivation

Several methods have been proposed in the graph neural network area. Most of them work by aggregating neighboring node features using a permutation invariant function (PMI). One of the most popular frameworks is the **graph convolution** which uses a PMI to aggregate features from neighboring nodes $n_j$ into a given node $n_i$ (See Appendix B). Another one is the **message-passing** that generates a message from each pair of nodes $\{n_i, n_j\}$ and aggregates them via a PMI. Furthermore, the **graph attention** computes the attention weight between $\{n_i, n_j\}$ and aggregates the neighboring features $n_j$ via a weighted sum of the attention weights.

In this work, we propose a different framework called multi-masked aggregators (MMA), where the network learns multiple weighted masks from pairs $\{u, v\}$ and aggregates them via a weighted PMI. Hence, the aggregation mechanism lies between graph attention which learns multiple masks, and message-passing, which uses invariant functions. Similarly to PNA [11], it benefits from increased expressivity from having multiple independent aggregators, and contrarily to GAT [3], it is not limited to a fixed number of heads during masking.

## 2.2   Flexible multi-aggregators

In recent work, it was demonstrated that using multiple uncorrelated aggregators during the message-passing increased the expressiveness while avoiding the exponential growth of the parameter space [11]. Their work proposed to use the *mean, max, min* and *std* operators to extract rich statistical features.

In this work, we build on the idea by using multiple learned aggregators that can also exhibit high-frequency filtering. We further combine the *mean, max, min* aggregators with multi-learned masks to provide a more expressive framework.

**Learning the Mask.**   The first step is to learn the mask $m_j^{l+1}$, with a unique value for each layer $l$ and pair of neighbouring nodes of $\{i, j\}$. To do so, we employ an MLP on the pair of node features $h_i, h_j$ and optionally the edge features $e_{ij}$ in a similar fashion to the MPNN. However, this does not constitute the message but rather the weights that will multiply the aggregated neighboring features. The equation is formalized in (1), with $\sigma$ being the activation function, $W_m$ a learned matrix for the $l$-th layer, and $||$ the column-wise concatenation.

$$m_j^{l+1} = MLP(||h_i^l, h_j^l, e_{ij}^l) = \sigma(W_m(||h_i^l, h_j^l, e_{ij}^l)) \tag{1}$$

In Equation (1), $m_j^{l+1}$ represents the learned mask of node j and $l$ represents the $l$th layer. Let $h_i^l, h_j^l$ and $e_{ij}^l$ be in $\mathbb{R}^N$, and then the concatenation of these vectors are in $\mathbb{R}^{3XN}$. $W_m$ is represented in $\mathbb{R}^{TX3}$. The multiplication of the concatenated $h_i^l, h_j^l$ and $e_{ij}^l$ with $W_m$ results in $\mathbb{R}^{TXN}$ dimension which gives the final dimension of $m_j^{l+1}$. $T$ represents the number of hidden units.

**Masked *Max/Min* Aggregators.**   *Max/Min* aggregators have shown to be effective for discrete tasks and domains where credit assignment and extrapolating to unseen distributions of graphs is important [14]. In this study, we extend max/min aggregators by adding a learned mask $m_j^l$. This allows the network to learn to ignore certain "undesired" nodes when propagating information.

$$max_i^l = max_{j \in N_i} (X_j^l * m_j^l) \qquad min_i^l = min_{j \in N_i} (X_j^l * m_j^l) \tag{2}$$

**Masked *Mean* Aggregator.**   One of the most widely used aggregators in the literature is the *mean* aggregator, in which each node computes a weighted average or sum of its incoming messages. Using a degree-scaler, it was also shown that the *sum* aggregation can be represented from the *mean* [11]. In this work, we first apply the same operation as in the LA-GCN [6] and then divide by the node's degree:

$$\mu_i(X^l) = \frac{1}{d_i} \sum_{j \in N_i} X_j^l * m_j^l \tag{3}$$

**Degree Scalers.**   In MMA, we further use degree scalers, motivated by their ability to amplify and attenuate signals using the node's degrees and increase expressivity [11]. The general equation is given below, with $S$ being the scaling factor, $d$ the node degree, $\alpha$ the amplification factor, and $delta$ the average degree in the training set. In our study, we use $\alpha = \{-1, 0, 1\}$, corresponding respectively to attenuation, no change, and amplification of the signal from its degree.

$$S(d, \alpha) = \left( \frac{\log(d+1)}{\delta} \right)^{\alpha}, d > 0, -1 < \alpha < 1 \tag{4}$$

**Combining Aggregators.** We further combine multiple aggregators and degree scalers to increase the expressivity of the network following the equation below. Here, $\otimes$ denotes the Tensor product and $\oplus_{mask}$ the general aggregation function of the proposed MMA framework.

$$\oplus_{mask} = \begin{pmatrix} I \\ S(D, \alpha = 1) \\ S(D, \alpha = -1) \end{pmatrix} \otimes \begin{pmatrix} Masked & Max \\ Masked & Min \\ Masked & Mean \end{pmatrix} \tag{5}$$

## 3 Experiments

We first evaluated the performance of MMA models on four widely-used benchmarking datasets (see Appendix A.1) over two tasks using a combination of different masked aggregators. Subsequently, we investigated the models' performances when the same type of masked aggregator(s) were used. Finally, the performance results of MMA were compared with the well-known baseline methods in the field. These methods are Message Passing Neural Networks (MPNN) [2], Graph Convolutional Networks (GCN) [1], GAT [3], LGCL [8], Graph-BERT [15], PNA [11], LA-GCN [6], Adaptive kernel graph neural network (AKGNN) [16], respectively.

### 3.1 Results

We trained several models using the multiple aggregator(s). Here, we used two different settings: In Setting 1 (see Table 3), we measured MMA's performance by combining different aggregators. In Setting-2 (see Table 5), the same type of aggregator(s) were used where each aggregator has a different trained mask. We used the same training/validation/test settings for a fair performance comparison with other methods. We also demonstrated some ablation studies in Appendix A.2.

Finally, we compared our best-performing results with the well-known baseline methods in the literature. The results are given in Table 1. Our results have shown improved performance over the compared methods in most cases.

**Table 1:** Benchmarking MMA on Pubmed, Citeseer, Cora and ZINC datasets. Detailed hyperparameter for MMA on each dataset can be found Table 4

| Models | Pubmed | Citeseer | Cora | ZINC |
|---|---|---|---|---|
| MPNN [2] | 75.60 | 64.00 | 78.00 | 0.288 |
| GCN [1] | 79.00 | 70.30 | 81.50 | – |
| GAT [3] | 79.00 | 72.50 | 83.00 | – |
| LGCL [8] | 79.50 | 73.00 | 83.30 | – |
| GRAPH-BERT [15] | 79.30 | 71.20 | 84.30 | – |
| PNA [11] | – | – | – | 0.188 |
| LA-GCN [6] | – | – | 81.50 | – |
| AKGNN [16] | 80.40 | 73.50 | 84.80 | – |
| MMA (ours) | **86.00** | **76.30** | **85.80** | **0.1562** |

## 4 Discussion and Conclusion

In this study, we propose Multi-Mask Aggregators for graph representation learning to utilize different and same aggregators within a learning mechanism. MMA provides a flexible learning method by integrating different or the same types of aggregator(s) where each has learnable parameters. Our contributions can be summarized as follows: **1)** It provides **flexible** multi-aggregators with the mask aggregation;**2)** It **unlocks** the limitation on the number of heads; **3)** It enables to extract **local informations** by local parameters instead of using global parameters like in MPNNs/PNA; **4)** It behaves between in **GAT** and **MPNN/PNA**; **5)** It increases the performance in **node classification** and **graph regression** benchmarks.

Besides all, as shown in the ablation studies, it was observed that there is no definite consensus on how much and which aggregator should be used. Authors believe that there is still room for improvement for aggregation fuctions in GNNs.

## Acknowledgements

The authors express their gratitude to Dominique Beaini for the valuable insights and discussions, as well as to Hacettepe University Biological Data Analysis Laboratory for the GPU support during the project.

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

# A Experiment Details

## A.1 Datasets

We trained our method on four different datasets: Cora, Citeseer, PubMed, and ZINC. The dataset statistics and the training, validation, and test settings are given in Table 2. The benchmarking datasets are explained below:

- **Cora** [17] - Cora is a citation graph dataset where each node represents a scientific publication classified as one of 7 classes. This dataset consists of 2,708 nodes and 5,429 edges, with an edge between two nodes if one cites the other. In this dataset, nodes are represented by binary feature vectors where each dimension indicates the absence or presence of the word from the dictionary containing 1,433 unique words. For the evaluation, the accuracy metric was used.

- **Citeseeer** [17] - Citeseer is also a citation graph dataset for node classification task where the nodes represent publications classified into six classes. Nodes are represented as binary feature vectors similar to the Cora dataset. In the Citeseer dataset, there are 3,327 nodes and 4,732 edges. For the evaluation, the accuracy metric was used.

- **Pubmed** [17] - Pubmed is another citation graph dataset where each node represents the papers related to diabetes. Pubmed is also a dataset for node classification where one of three classes is assigned to each node. This dataset consists of 19,717 nodes and 44,338 edges. Here, each node is represented by a feature vector that shows TF/IDF weighted word vector from the dictionary with 500 unique words. For the evaluation, the accuracy metric was used.

- **ZINC** [18] - ZINC is a graph regression dataset for constrained solubility prediction of chemical compounds. In this dataset, each compound is represented by a graph where nodes represent atoms and edges represent the bonds between atoms. The ZINC dataset consists of 12,000 molecules with varying atom numbers from 9 to 37. The mean absolute error (MAE) metric was used for the evaluation.

The dataset statistics are summarized in Table 2.

**Table 2:** Summary of the datasets used in benchmarking

| Domain & Construction | Dataset | #Nodes | Total #Nodes | Edges | Features | Classes | Train/Val./Tes | Task |
|---|---|---|---|---|---|---|---|---|
| **Social Networks:** Real-world citation graphs | Cora | 2,708 | 2,708 | 5,429 | 1,433 | 7 | 1,208/500/1,000 | Node Classification |
| **Social Networks:** Real-world citation graphs | Citeseer | 3,327 | 3,327 | 4,732 | 3,703 | 6 | 1,827/500/1,000 | Node Classification |
| **Social Networks:** Real-world citation graphs | Pubmed | 19,717 | 19,717 | 44,338 | 500 | 3 | 18,217/500/1,000 | Node Classification |
| **Chemistry:** Real-world molecular graphs | ZINC | 9-37 | 277,864 | – | – | – | 10,000/1,000/1,000 | Graph Regression |

## A.2 Ablation Studies

Table 3 shows the performance results of the top four best-performing models trained using different combinations of multi-mask aggregators. As it can be observed from Table 3, there is no consensus on types of aggregators when we consider the performance results based on different datasets; therefore, dataset-specific aggregators should be determined empirically. For example, Masked Mean-Mean2 aggregators performed best on the Cora dataset, whereas Masked Min-Min2-Min3 aggregators worked best on the Citeseer dataset. Similarly, using Min-Min2-Min3-Min4 and Min-Max sets of aggregators resulted in better results in Pubmed and ZINC databases, respectively.

We also evaluated the performance of the MMA using the same multi-masked aggregators. Here, MMA models were trained using single or multiple types of the same aggregators to investigate how models' performances change with the same type of masked aggregator(s). The results are shown in Table 5. Here, we trained MMA models using up to 4 same aggregators with different learnable masks. When the results are investigated, it can be seen that specific single multi-aggregators work better than the remaining aggregators on different datasets. For example, on the Cora dataset mean aggregators almost consistently worked better than the min and max aggregators. Considering the Citeseer dataset, we can see that max aggregators work with less performance than the min and mean aggregators.

**Table 3:** Performance of MMA using different aggregators on independent test sets (Setting-1)

| Dataset | Masked Aggregators | Learning Rate | Weight Decay | Hidden Units | Epoch | Accuracy/MAE |
|---|---|---|---|---|---|---|
| **Cora** | Mean-Min | 0.001 | 5e-4 | 128 | 200 | **85.10** |
| | Mean-Max-Min | 0.001 | 3e-4 | 128 | 1000 | 84.30 |
| | Mean-Max | 0.001 | 1e-4 | 128 | 1000 | 84.10 |
| | Max-Min | 0.01 | 3e-4 | 64 | 1000 | 83.60 |
| **Citeseer** | Mean-Max | 0.01 | 5e-4 | 128 | 500 | **75.90** |
| | Mean-Min | 0.01 | 3e-4 | 64 | 500 | 75.50 |
| | Max-Min | 0.001 | 1e-4 | 64 | 1000 | 75.30 |
| | Mean-Max-Min | 0.01 | 5e-4 | 64 | 500 | 75.30 |
| **Pubmed** | Mean-Min | 0.01 | 1e-4 | 64 | 200 | **85.90** |
| | Mean-Max-Min | – | – | – | – | – |
| | Mean-Max | – | – | – | – | – |
| | Max-Min | – | – | – | – | – |
| **ZINC** | Mean-Min | 0.0001 | 3e-4 | | 10000 | 0.1585 |
| | Mean-Max-Min | 0.00001 | 3e-4 | – | 10000 | 0.1606 |
| | Mean-Max | 0.0001 | 3e-4 | – | 10000 | 0.1585 |
| | Min-Max | 0.0001 | 3e-4 | – | 10000 | **0.1562** |

**Table 4:** Detailed hyperparameter for best performance of MMA in Table 1

| Dataset | Masked Aggregators | Learning Rate | Weight Decay | Hidden Units | Epoch | Accuracy/MAE |
|---|---|---|---|---|---|---|
| **Cora** | Mean-Mean2 | 0.001 | 3e-4 | 64 | 200 | 85.80 |
| **Citeseer** | Min-Min2-Min3 | 0.01 | 3e-4 | 128 | 500 | 76.30 |
| **Pubmed** | Min-Min2-Min3-Min4 | 0.01 | 5e-4 | 16 | 500 | 86.00 |
| **ZINC** | Min-Max | 0.0001 | 3e-4 | – | 10000 | 0.1562 |

# B    Theoretical Background

Due to the lack of order in most real graphs, permutation invariance is an essential feature for aggregation functions. While aggregating representations of the node's neighbors, the neighborhood aggregation scheme iteratively updates the representation of a node [6]. This intuition explained for the aggregation process can be formalized as follows:

$$s_i^{(k-1)} = f_{ag}^{(k)}(h_j^{(k-1)}, j \in N_i) \tag{6}$$

where $f_{ag}^{(k)}$ is aggregator in the k-th layer. The aggregation function $f_{ag}^{(k)}$ should be a permutation invariant function on a multiset. According to [19], definition of permutation invariant function described as:

**Definition 1:** A function f is permutation-invariant if

$$f(h_1, h_2, ..., h_{|N_i|}) = f(h_{\pi_{(1)}}, h_{\pi_{(2)}}, ..., h_{\pi_{(|N_i|)}}) \tag{7}$$

for any permutation $\pi$ and $|N_i|$ is the length of the sequence. $\Pi_{|N_i|}$ denotes the multiset of all permutations of the integers 1 to $|N_i|$ and $h_\pi, \pi \in \Pi_{|N_i|}$, denotes a reordering of the multiset according to $\pi$. The relation between set and permutation invariant function can be shown in the following theorem in [19]:

**Theorem 1:** A function operating on a multiset $h_1, h_2, ..., h_{(|N_i|)}$ having elements from a countable universe is a valid set function. It is invariant to the permutation of instances in the multiset if it can be decomposed in the form $\rho(\sum_{\pi \in \Pi_{|N_i|}} \phi(h_\pi))$ for suitable transformation $\phi$ and $\rho$.

Theorem 1, it can be inferred that all the representations are added and then applied to nonlinear transformation.

Mean, sum aggregation functions and aggregators in GCN and GAT can be represented in this concept. As shown in Eq.(8) and Eq.(9), respectively, GCN and GAT add up all neighborhood with fixed parameters or learnable parameters.

**Table 5:** Performance results of same multi-masked aggregators on independents test sets (Setting-2)

| Masked Aggregators | Cora | Citeseer | Pubmed | ZINC |
|---|---|---|---|---|
| Mean | 85.60 | 76.00 | – | 0.1631 |
| Mean-Mean2 | **85.80** | 76.10 | – | 0.1763 |
| Mean-Mean2-Mean3 | 84.60 | 74.60 | – | 0.1940 |
| Mean-Mean2-Mean3-Mean4 | 84.80 | 75.20 | – | 0.1886 |
| Min | 83.90 | 76.10 | 85.80 | 0.1535 |
| Min-Min2 | 84.20 | 75.40 | 85.30 | 0.1571 |
| Min-Min2-Min3 | 84.00 | **76.30** | 85.70 | 0.1591 |
| Min-Min2-Min3-Min4 | 84.00 | 75.70 | **86.00** | – |
| Max | 83.60 | 75.40 | 85.50 | **0.1519** |
| Max-Max2 | 83.00 | 75.00 | 84.30 | 0.1653 |
| Max-Max2-Max3 | 83.00 | 75.00 | 83.30 | 0.1717 |
| Max-Max2-Max3-Max4 | 83.60 | 75.00 | 81.90 | 0.1604 |

$$s_i^{(k-1)} = f_{agg}^{(k)}(h_j^{(k-1)}) = \sum_{j \in N_i} h_j^{(k-1)} / \sqrt{d_i d_j} \tag{8}$$

where $d_i$ and $d_j$ are the node degrees of node $v_i$ and node $v_j$ respectively.

$$s_i^{(k-1)} = f_{aga}^{(k)}(h_j^{(k-1)}) = \sum_{j \in N_i} \alpha_{ij} h_j^{(k-1)} \tag{9}$$

where $\alpha_{ij}$ is a learnable attention coefficient that indicates the importance of $v_j$ to $v_i$.

### B.1 Mask Aggregator

[6] tried to use mask aggregator with sum aggregation function to assign different importance to different neighbor's features. In this aggregation process, they tried to use a mask aggregator, which assigns different weights to different neighbor's features and then aggregates by sum aggregation function. It can be shown as the following:

$$s_i^{(k-1)} = f_{agm}^{(k)}(h_j^{(k-1)}) = \sum_{j \in N_i} h_j^{(k-1)} * m_j^{(k-1)} \tag{10}$$

where $h_j^{(k-1)} \in \mathbb{R}^{d_{k-1}}$, $m_j^{(k-1)} \in \mathbb{R}^{d_{k-1}}$ is a specific mask for each neighbor, produced by the auxiliary model. They showed that the mask aggregator is permutation invariant as the following theorem, which is proven by [15]:

**Theorem 2:** $f_{agm}^{(k)}$ is a permutation-invariant function acting on finite but arbitrary length sequence $h_j^{(k-1)}, j \in N_i$.

**Proof 2:** Given $H = h_1^{(k-1)}, h_2^{(k-1)}, ..., h_{(|N_i|)}^{(k-1)}$ a finite multiset, and $h_j^{(k-1)} \in \mathbb{R}^{d_{k-1}}$, mask aggregator was tried to be combined with a fixed output $s_i^{(k-1)} \in \mathbb{R}^{d_{k-1}}$ as follows:

$$s_i^{(k-1)} = f_{agm}^{(k)}(h_j^{(k-1)}) = \sum_{j \in N_i} h_j^{(k-1)} * m_j^{(k-1)} \tag{11}$$

where $m_j^{(k-1)} \in \mathbb{R}^{d_{k-1}}$ is a specific mask for each neighbor produced by the auxiliary model. First, it was tried to get mask $m_j^{(k-1)}$ for each node $h_j^{(k-1)}$ by using an auxiliary model given graph information.

There exists a mapping function $\phi : \mathbb{R}^{d_{k-1}} \to \mathbb{R}^{d_{k-1}}$ that can formulate $h_j^{(k-1)} * m_j^{(k-1)}$ to $\phi(h_j^{(k-1)})$, and (11) can be written as:

$$s_i^{(k-1)} = f_{agm}^{(k)}(h_j^{(k-1)}) = \sum_{j \in N_i} \phi(h_j^{(k-1)}) \tag{12}$$

and $\rho$ can be seen as $\rho = 1$. (8) can be seen as a permutation of H, according to [19].

Next, they prove there exist an injective mapping function $\phi$, so that $f_{agm}^{(k)}(h_j^{(k-1)})$ is unique for each finite multiset H.

Since H is countable, each $(h_j^{(k-1)}) \in H$ can be mapped to a unique element to prime numbers $p(H) : \mathbb{R}^M \to \mathbb{P}$ by a function $p(h_j^{(k-1)})$. $\phi(h_j^{(k-1)})$ can be represented as $-log p(h_j^{(k-1)})$. Thus,

$$f_{agm}^{(k)}(h_j^{(k-1)}) = \sum_{j \in N_i} \phi(h_j^{(k-1)}) = log p(h_j^{(k-1)}) \tag{13}$$

takes a unique value for each distinct H.

Besides, the dimension $d_{d-1}$ of the latent space should be at least as large as the maximum number of input elements $|N_i|$, which is both necessary and sufficient for continuous permutation-invariant functions [20].

Since a neural network can approximate any continuous function, according to the universal approximation theorem [21], MLPs can be used as an auxiliary model and learn $\phi$ and $\rho = 1$.

## B.2 Multi Aggregator

According to Theorem 2, it can be concluded that multi-set is a permutation-invariant function, and mask aggregator can adapt which features or neighbors are essential and filter the noisy information.

However, [11] showed that sum aggregation does not discriminate between graphs, and they proposed multi-aggregation to tackle this problem. They showed that the multi-aggregation can discriminate between graphs as the following theorem and proof:

**Theorem 3:** In order to discriminate between multisets of size n whose underlying set is $R$, at least $n$ aggregators are needed.

Unlike the [5], [11] consider a continuous input feature space; this better represents many real-world tasks where the observed values have uncertainty and better models the latent node features within a neural network's representations. Continuous features make the space uncountable and void the injectivity proof of the sum aggregation presented by Xu et al. [5]. Hence, they redefine aggregators as continuous functions of multisets that compute a statistic on the neighboring nodes, such as mean, max, or standard deviation. Continuity is important with continuous input spaces, as small variations in the input, should result in small variations of the aggregators' output.

