# OpenReview forum: "Multi-Mask Aggregators for Graph Neural Networks"
_logconference.io/LOG/2022/Conference — LoG 2022 Poster_

### Official Review · Reviewer_dCgK · 2022-09-30

**Overall Score:** 6
**Confidence:** 3

**Review:**

This paper proposes a novel aggregation operator for GNN: Multi-Mask Aggregators (MMA). It learns a weighted mask for each type of aggregation operator and combines the outputs into the new node representation. The improvement over baselines is very significant and impressive.

I don't think there exists any major weakness although the proposed method is straightforward and the performance gain is possibly due to the introduced additional parameters compared to baseline.

---

### Official Review · Reviewer_h1f7 · 2022-10-17

**Overall Score:** 6
**Confidence:** 3

**Review:**

**1.  Summarise the main contribution of this paper.**

This paper proposed a multi-mask neighboring feature aggregator to fuse features for one node. It builds on Gabriele Corso et al. [1] that mean/max/min/std operators give expressive and rich features, the authors have proposed to learn multiple mask for Mean/Min/Max operator aggregators separately. So that the aggregated feature is more expressive than existing GAT, GCN like features, and gives better performance than those comparing methods on 4 datasets. I think the contribution lies in two main sides. 1. integrate mean/min/max fusing operator in one neural network. 2. the fusing strategy is tunable and adjustable in a data-driven way.

**2. Strong points shown in this paper:**

* Detailed analysis and even theoretical proof of their motivation of their multi-mask aggregators design.
* Thorough experimental discussion, which makes this paper convincing and reasonable.
* The writing is clear.
* The authors shared their source code.

**3. Weak points in this paper:**

Although there are some advantages delivered in this paper, I still have several concerns:

*  **Lack of Implementation Detail**, I can see some basic details about model training/implementation detail in Table 3. But where is the detailed implementation detail about the result reported in Table 1? Since Table 1 is a big table that contributes most to the paper idea, I could not find any discussion about it.

* **Lack of More Discussion**, the performance improvement shown in this paper owes to three MLP-learned aggregators. Adding extra MLP have equally added extra parameters to the whole model. So one question in my mind is that: **Does the performance improvement owe more to the extra introduced parameters than the mean/min/max aggregators?**, To make the methods discussed in the paper more convincing, **the author may need to do an ablation study** in which just directly applying mean/min/max operators to aggregate features, to see how the performance goes. Such ablation study also helps to show if jointly using mean/min/max operators truly help the graph neural network learning.

4 **Recommendation**

* I am currently holding the **"borderline"** recommendation. I personally like the "multi-mask" idea but, as I have shown in the weak-points section, there are some strong concerns I am curious about. I prefer to wait for the authors' feedback first to decide if I would change my rating.

5 **Type of the Paper**.

Based on the writing in the paper, I think it is a 4-page track. (considering the fifth page is Reference)

[1] Gabriele Corso et al. Principal Neighbourhood Aggregation for Graph Nets. NeruIPS 2020.

---

### Official Review · Reviewer_hDPi · 2022-10-17

**Overall Score:** 5
**Confidence:** 4

**Review:**

Summary

This work proposes a novel model MMA where the network learns multiple weighted masks from the node pair and aggregate them with some different of the same type of aggregates.

Advantage

The model leverages the increased expressivity from the multi-aggregator models such as PNA and the learnable masks from LA-GCN.
According to the result of the experiments, the performance of the model MMA is better, compared with LA-GCN and LGCL.

Disadvantage

There aren't enough details about the MMA model. For example, there should be more information about the masked aggregator in eq(1) such as the shape of m_{j}^{l}.

In Table 1, the performance of MMA is the worst on the dataset ZINC, compared with MPNN and PNA. Could you give more explanations?

There should be more details about the experiments, such as the setting of the parameters. In fact, in the experiments, the author didn't even clarify the task in detail or the metrics to measure the performance of MMA.

In table 1, there are only the performances of MPNN, PNA, and MMA on the dataset ZINC, and this paper didn't give the reason.

There are many grammar mistakes, increasing the difficulty of reading.

---

### Official Review · Reviewer_YwqP · 2022-10-19

**Overall Score:** 8
**Confidence:** 5

**Review:**

Summary Of the Paper:

The paper proposed a method of Multi-Mask Aggregators (MMA) to learn a weighted mask for each aggregator before collecting neighboring messages. Intuitively, the MMA framework is not limited by the number of heads from GAT and has more discriminative than an MPNN.

Overall, I think the novelty of this work is slightly incremental and empirical. However, as an extended abstract, I give this work “Weak Accept”. Here are the strengths, weaknesses, question, and suggestion of this work:

Strengths of the paper:

(1). This work is still enlightening, although some slightly incremental and empirical study is therein.

(2). The sentences of this paper are well-written.

Weaknesses of the paper:

(1). The number of experimental datasets is insufficient. The experiments are recommended to add some other datasets.

(2). This work uses multiple MLPs to learn masks. In this learning process, $h_i^l$, $h_j^l$, and $e_{ij}^{l}$ vectors are concatenated. Following this column-wise concatenation, I worry about the training efficiency when the dimension of such vectors is large. Maybe these large vectors make the model out of memory.

Question of the paper:

(1). Do the authors consider the efficiency of model training? Especially when the eigenvector dimension is particularly large.

Suggestion for the paper:

(1). Maybe the length of this extended abstract limits the idea of this work, making it slightly incremental and empirical. I very much expect the authors to extend the idea to the full paper by using solid theorems and efficient-and-feasible training strategies. By doing so, this work will be more solid and perfect.

----------------------------------------- new comments ---------------------------------------------

The authors answered my questions sufficiently, so I decided to raise the overall score of this work. I expect that the authors develop this work in the future, especially in the efficient training of Multi Mask Aggregators.

---

### Meta-Review · Area_Chair_GhDz · 2022-11-16

**Confidence:** 5
**Recommendation:** Accept

**Meta Review:**

The authors propose Multi-Mask-Aggregators (MMA), in which weighted masks are learned across multiple aggregators in a GNN to obtain new node representations, and thus can be seen as a combination of multi-aggregator models such as PNA and the learnable masks from LA-GCN. Despite the incremental contribution, the experimental evaluation reveals that MMA is able to out-perform both methods by a large margin.

Most reviewers are in favor of accepting this work. Overall, I share similar concerns as the reviewers, such as the insufficient number of experimental datasets being used (with all of them being also small-scale), missing ablation studies (e.g., which aggregator benefits the most from masking), and missing details of the underlying model. In all fairness though, it is hard to incorporate all of this in an extended abstract, so I also recommend acceptance. I strongly recommend the authors to take the feedback from reviewers into account when working on a full paper submission.

---

### Decision · Program_Chairs · 2022-11-23

Accept (Poster)